# Self-reported health, neuropsychological tests and biomarkers in fully recovered COVID-19 patients vs patients with post-COVID cognitive symptoms: A pilot study

Michael R. Lawrence[1,2], Judith E. Arnetz [ID][3]*, Scott E. Counts[3,4], Aiesha Ahmed[2], Bengt B. Arnetz [ID][3]

**1** Division of Clinical Neuropsychology, Corewell Health, Grand Rapids, Michigan, United States of America, **2** Department of Neurology, Corewell Health, Grand Rapids, Michigan, United States of America, **3** Department of Family Medicine, College of Human Medicine, Grand Rapids, Michigan, United States of America, **4** Department of Translational Neuroscience, College of Human Medicine, Grand Rapids, Michigan, United States of America

* arnetzju@msu.edu

## Abstract

Substantial numbers of individuals who contract COVID-19 experience long-lasting cognitive symptoms such as brain fog. Yet research to date has not compared these patients with healthy controls with a history of laboratory-confirmed COVID-19 infection, making it difficult to understand why certain COVID patients develop post-COVID cognitive symptoms while others do not. The objective of this pilot study was to compare two groups of laboratory-confirmed post-COVID patients, with and without cognitive symptoms, on measures of cognitive and psychological functioning, self-reported perceptions of functional status and quality of life, and biomarkers of stress, inflammation, and neuroplasticity. Using a case-control design, 17 participants were recruited from a healthcare system in western Michigan, USA in 2022–2024. All participants were aged 25–65 and had a positive polymerase chain reaction (PCR) test confirming previous COVID-19 infection. Ten participants reported cognitive symptoms (long COVID group) while seven were fully recovered with no residual symptoms (controls). All participants underwent an interview on their self-rated health and quality of life, a battery of neurocognitive tests, and blood draw for biomarker analysis. No group differences were detected for neuropsychological test measures except for letter fluency where the long COVID group scored significantly lower (p < .05). The long COVID group had significantly lower ratings than controls on quality of life, physical health, emotional functioning, and psychological well-being. Serum levels of nerve growth factor (NGF), a biomarker of brain plasticity, were significantly lower in the long COVID group, which was significantly more likely than controls to have serum levels of inflammatory marker (interleukin (IL)-10) values greater than or equal to the median (p = 0.015). Biomarker analyses suggest possible prolonged

**Data availability statement:** All relevant data are within the paper and its Supporting Information files.

**Funding:** This study was financially supported by a grant from the Spectrum Health Foundation, principal investigators AA and BBA. The funders did not play any role in the study design, data collection and analysis, decision to publish, or preparation of the manuscript.

**Competing interests:** The authors have declared that no competing interests exist.

inflammatory processes in long COVID patients compared to fully recovered patients. Results of decreased neuroplastic functioning give credence to patients' reports of post-COVID changes in brain function.

## Introduction

Post COVID conditions (PCC), or long COVID, is a clinical syndrome characterized by symptoms of COVID-19 that extend well beyond the initial recovery period, a minimum of 12 weeks and often for a year or more [1]. The clinical diagnosis, termed post-acute sequela of COVID-19 (PASC), was officially defined and assigned in the International Classification of Diseases (ICD) - 10 in October of 2021. However, a widely accepted case definition and associated symptom timeframe is still under development [2]. Research suggests that approximately 30% of individuals who contract COVID-19 will go on to develop a post COVID-19 condition and experience long-lasting symptoms [3]. Although symptoms are often diverse, many patients experience persistent cognitive complaints, including poor memory and "brain fog" [1,4]. To date, five years after the onset of the COVID-19 pandemic, cognitive symptoms continue to plague a substantial proportion of individuals who have experienced the viral infection [5].

A large survey of U.S. adults conducted during a six-month period in 2021–2022 found that 46% of those with PCC reported either brain fog or impaired memory, which was associated with a decreased likelihood of working full-time [6]. Another U.S. survey of nearly 15,000 adults with prior COVID-19 infection found that 57% of individuals with PCC experienced significantly more cognitive symptoms at least daily compared to those without PCC [7]. The most commonly reported cognitive disturbances involved attention, executive functioning and memory, with brain fog or processing speed deficits reported in 40–60% of patients with long COVID [8]. Brain fog, or COVID fog [9] is often accompanied by fatigue, anxiety, and depressive symptoms similar to those seen in chronic fatigue syndrome [10]. Post-COVID cognitive symptoms are reported more frequently among individuals who had a mild initial infection [10–12] and in individuals who report prior cognitive difficulties and a diagnosis of depressive disorder [13]. Individuals with an elevated psychological symptom burden as part of their post COVID-19 clinical picture report more perceived cognitive difficulties in day-to-day life [14]. Post-COVID cognitive symptoms have also been associated with decreased quality of life and function [15].

Notably, research to date has not shown significant objective cognitive deficits in individuals with cognitive complaints related to long COVID. A study of 51 adults with post-COVID cognitive symptoms reported that they scored in the normal range on standardized neuropsychological measures. However, that study lacked a control group [14]. The results of neuropsychological tests in 53 outpatients diagnosed with COVID-19 revealed no scores in the impaired range, leading the authors to conclude that objective cognitive performance was not affected by self-reported cognitive complaints [16]. One study compared adults with post-COVID conditions (n = 319) to healthy controls (n = 109) and found no significant group differences in

neuropsychological test results. Healthy controls in that study had not been infected with COVID-19. However, when dividing the PCC group into those with cognitive complaints (n = 123) and those without (n = 196), they found that those with cognitive complaints scored significantly worse on global cognition, learning, memory, processing speed, language, and executive function [17].

A number of studies [e.g.,18–24] have focused on biomarkers associated with post-COVID symptoms in an effort to better understand PCC, improve diagnosis, and develop potential therapeutic treatments. Lai et al. [22] identified interleukin (IL)-6, C-reactive protein (CRP), tumor necrosis factor alpha (TNF-alpha), and neurofilament light chain (NfL) as potential diagnostic indicators of long COVID. In a Brazilian cohort, Queiroz et al. 23] found that patients with long COVID exhibited higher levels of IL-17 and IL-2, whereas patients with no post-COVID symptoms had higher levels of IL-4, IL-6 and IL-10. While patients in that study had all been diagnosed with COVID, the authors only compared biomarkers between groups and did not report on cognitive complaints. However, due to the broad array of symptoms associated with PCC, as well as the varied study designs employed, it is difficult to draw any definitive conclusions regarding specific biomarkers. A few studies have focused more specifically on biomarkers associated with cognitive impairment [25,26]. In a study of 710 COVID survivors, Damiano et al. [25] found no cytokines or inflammatory markers associated with cognitive performance. Vrettou et al. [26] studied neural biomarkers in a cohort of 65 long COVID patients and 29 age and sex-matched healthy controls with no known history of COVID infection. They found that levels of glial fibrillary acidic protein (GFAP), a biomarker of neural inflammation, were significantly higher among long COVID patients, but were not correlated with the presence of long COVID symptoms.

In summary, the literature to date offers sparse evidence of biomarker response to post-COVID cognitive complaints, and we lack standardized phenotyping for biomarker analyses [27]. Moreover, patients with PCC cognitive complaints scored lower than healthy controls on neurocognitive testing but did not score worse than PCC patients without cognitive complaints [17]. The authors concluded that all patients with PCC may experience cognitive difficulties, even though they may not express cognitive complaints. However, patients with PCC cognitive complaints may also differ in terms of psychological symptoms and self-perceived health and functional status [14,26,28]. In any event, large numbers of individuals with PCC continue to suffer from cognitive symptoms that do not seem to improve with time [5]. Studies to date have compared cognitive issues between COVID laboratory-positive and laboratory-negative patients [28,29] or followed longitudinal cohorts of lab-confirmed COVID patients without comparison groups [12,13]. The lack of studies involving controls with a history of laboratory-confirmed COVID-19 infection makes it difficult to enhance our understanding of why certain COVID patients develop PCC cognitive symptoms while others do not. To address these gaps in the literature, the aim of the present pilot study was to compare two groups of laboratory confirmed post-COVID patients, with and without cognitive symptoms, on measures of cognitive and psychological functioning, self-reported perceptions of functional status and quality of life, and biomarkers of stress, inflammation, and neuroplasticity.

## Methods

### Study design

This was a pilot study utilizing a case-control design. Investigators were not blinded as to participant group (case or control).

### Study participants and recruitment

All participants were recruited from Corewell Health medical facilities in western Michigan, U.S. Inclusion criteria included being between the ages of 25–65, at least 6 months after COVID infection, and a positive polymerase chain reaction (PCR) test confirming COVID infection. All participants were placed into one of two groups, a symptomatic group based upon self-report of cognitive symptoms ("long COVID"), or an asymptomatic control group. The asymptomatic group was fully recovered from COVID, with no residual symptoms. Attempts were made to age and sex match the symptomatic

and control group. Patients with residual symptoms (cases) were identified by neurological medical providers and referred for neurocognitive evaluation. Controls were identified via electronic health records as patients who had sought care for COVID-19 at the hospital, as well as via posters in the hospital. Each participant underwent a brief interview assessing symptom complex and illness time frame/course. Participants were asked to endorse (yes/no) whether they were currently experiencing attention deficits, breathing difficulties, changes in mood, difficulties problem solving, fatigue, loss of smell, loss of memory, mental fogginess, muscle aches, paresthesia/pins and needles, slow processing speed, or other symptoms. We did not assess for preexisting mood disturbances, although we did assess for change in mood. The rationale was that there is a high base rate of mood disturbance in the general population [30,31]. Additionally some research suggests that those with preexisting mood disturbance experience long COVID at a higher rate [32]. Thus, ruling out all pre-existing mood disturbance would actually restrict the range too much and not represent a valid clinical sample.

Of note, inclusion criteria were based on our main focus on cognitive symptoms among working-age patients with PCC. Varying definitions of long COVID have been established by the World Health Organization (WHO) [33], Centers for Disease Control and Prevention (CDC) [34], National Academies of Sciences, Engineering and Medicine (NASEM) [35], and National Institute for Health and Care Excellence (NICE) [36]. All of these define long COVID as characterized by a broad array of symptoms that are present at least 3 months after the onset of SARS CoV-2 infection and that are not explained by alternative diagnoses. The WHO definition explicitly includes cognitive dysfunction as a symptom [33]. In the current study, we chose a longer period of 6 months post COVID infection to ensure the presence of residual cognitive symptoms in the long COVID patients, as well as the absence of cognitive symptoms in the controls. This is in line with a global systematic literature review that found that the majority of studies on long COVID have not used the definitions provided by these major public health organizations. Instead, due to the specific nature of each study, authors establish their own definitions or refer to other authors' definitions [37]. This underscores the difficulty in establishing a standard definition of a complex condition.

Exclusion criteria included adults who were unable to consent, pregnant women, prisoners, and anyone who reported having received monoclonal antibody treatment or COVID convalescent plasma for COVID-19. Additionally, participants with preexisting neurologic illness (i.e., stroke, tumor, head injury, or preexisting cognitive complaints) were also excluded. Vaccination status was not an inclusion or exclusion criterion. However, vaccination status and dates of vaccination were recorded for all cases and controls.

## Sample size

This was an exploratory pilot study that was planned as the first step towards a larger-scale investigation. Based on a rate of 1–2 COVID survivor patients being seen in the Neurology clinic per week, the team expected to achieve recruitment of 15 cases within 3–6 months. Recruitment of controls was expected to take somewhat longer. Using a flat rule of thumb [38], a sample size of 15 cases and 15 controls was considered the minimum to detect clinically significant differences between groups in neurocognitive measures and biomarkers. While formal power calculations are not required for pilot studies [38], it was important to have a sufficient sample size to examine the feasibility of recruiting participants with and without long COVID, and whether differences could be detected between groups [39].

## Data collection

Recruitment took place between March 4, 2022 and April 10, 2024. Each participant met with a study coordinator who verbally explained the nature of the study, the research intent, and risks. Participants then read and signed informed consent forms prior to participation. All participants underwent an interview on their self-rated health and quality of life, a battery of neurocognitive tests, and blood draw for biomarker analysis. Each participant was reimbursed $75 upon completion of the approximately 2 - hour evaluation.

## Self-report symptoms

All participants completed a brief structured interview assessing symptom complex and time frame/course. The self-report measures focused specifically on mood, emotional functioning and quality of life. Mood was measured using the Beck Depression Inventory [40] and Beck Anxiety Inventory [41]. Emotional functioning and quality of life were measured using the Short Form Health Survey SF-36 [42,43] and the EuroQuol 5-Dimension EQ-5D [44]. Descriptions of each measure are provided in Table 1.

## Neurocognitive testing

The interview was followed by a 90-minute neurocognitive battery assessing various aspects of cognition including estimated premorbid functioning, attention, processing speed, verbal fluency, learning and memory, visual planning, and executive functioning (e.g., problem-solving, multitasking, sustained concentration). An overview and description of the tests is provided in Table 2. All measures used in the current study were well known for their strong psychometric properties.

## Biomarker analysis

Blood serum and plasma specimens were taken for biomarker analysis. All blood sampling was conducted at the health system laboratory. For each participant, 30 mL blood was drawn by a licensed phlebotomist, and the time of day of sample collection was recorded. All samples were frozen and stored in a –80 degree C freezer for later analysis. Analysis focused on biomarkers of stress (cortisol), anti-stress or recovery (dehydroepiandrosterone sulphate, DHEA-S), inflammation (IL-6, IL-10, TNF-alpha), neuroplasticity (brain-derived neurotrophic factor, BDNF, and nerve growth factor, NGF) and neurode-generative disease (neurofilament light chain, NfL, pTau 181, pTau 217). The inflammatory markers selected have also been associated with poor cognitive function [53–55]. Analyte concentrations in both serum and plasma samples were measured using human-specific 96 well ELISAs with the exception of NfL, which was analyzed by Single Molecule Array (SIMOA) technology. The following kits were used to generate data: Cortisol (Boster #EK7002, Pleasanton, CA; sensitivity > 20 ng/ml), DHEA-S (Invitrogen #EIAD HEA, Carlsbad, CA; > 90.9 ng/ml), IL-6 (BioLegend #430507, San Diego, CA; > 1.6 pg/ml), IL-10 (BioLegend #430607, San Diego, CA; > 2 pg/ml), BDNF (Origene #EA100205, Rockville, MD; > 2 pg/ml), NGF (AVIVA #OKEH00186, San Diego, CA; > 15.6 pg/ml), and NfL (Quanterix #193186, Billerica, MA; > 1.38 pg/

**Table 1. Self-Report Questionnaires.**

| Report Inventory | Description |
|---|---|
| *Mood* | |
| **Beck Depression Inventory** [40] | 21-item self-report inventory measuring attitudes and symptoms of depression. Each item is rated on a scale from 0 to 3, with 0 indicating no symptoms and 3 indicating strong symptoms. |
| **Beck Anxiety Inventory** [41] | 21-item self-report inventory measuring anxiety level and severity. Items are rated on a scale from 0 (not at all) to 3 (severely – it bothered me a lot). |
| *Emotional Functioning/Quality of Life* | |
| **Short-Form Health Survey (SF-36)** [42,43] | 36-item survey measuring 8 domains of health: (1) Limitations in physical activities due to health problems; (2) Limitations in social activities due to physical or emotional problems; (3) Limitations in usual role activities due to physical health problems; (4) Bodily pain; (5) General mental health; (6) Limitations in usual role activities due to emotional problems; (7) Vitality; (8) General health perceptions. |
| **EuroQol 5-Dimension (EQ-5D)** [44] | Patient reported outcome that measures quality of life across 5 domains: mobility, self-care, usual activities, pain/discomfort, and anxiety/depression. Each dimension is scored on a 3-level severity ranking that ranges from "no problems" through "extreme problems." Also includes a 0–100 visual analogue scale to assess patient's self-rated overall health (100 = the best health you can imagine; 0 = the worst health you can imagine). |

**Table 2. Neuropsychological Test Measures.**

| Test Measure | Test Description |
|---|---|
| *Premorbid Intellect* | |
| **Test of Premorbid Functioning** [45] | Estimate of premorbid intellect as measured by single word reading ability |
| *Attention* | |
| **Wechsler Adult Intelligence Scale Digit Span (WAIS-IV Digit Span)** [46] | Auditory attention task assessing working memory capacity; the forward memory span involves repeating sequence of numbers verbatim, while the backward memory span involves recalling numbers in reverse order. A sequencing span task is also administered in which participant orders numbers from lowest to highest. |
| *Processing Speed* | |
| **Symbol Digit Modalities Test (SDMT)** [47] | Digit-symbol transcription task involving selective attention, visual scanning, and cognitive speed; Using a reference key, participant has 90 seconds to pair specific numbers with given geometric figures. Responses can be written or given orally. |
| *Language* | |
| **Letter fluency (COWAT)** [48] | Speeded word retrieval based on letter cues; performance is based on total number of correct responses. |
| **Semantic Fluency (Animal fluency)** [48] | Rapid word generation for a semantic category; performance is based on total number of correct responses. |
| *Visual Planning* | |
| **Rey-Osterrieth Complex Figure Test** [49] | Figure copy task assessing visuo-constructional ability as well as planning/organization. |
| *Learning and Memory* | |
| **California Verbal Learning Test (CVLT-II)** [50] | Measure of learning and recall of a word list; after the initial learning phase, participants are asked to recall the list after a 20-minute delay. Indices examined include total recall (trials 1–5), free/cued short-delay recall, and free/cued long delay free recall. |
| *Executive Functioning* | |
| **Trail Making Test** [51] | Part A involves visual search and speed; participants draw a continuous line through 25 consecutively numbered circles. Part B introduces added cognitive demand of mental set-shifting; participant draws a continuous line through consecutive numbers and letters, alternating the order each time a connection is made (1-A-2-B-3-C, etc.). |
| **Tower of London (TOL-2)** [52] | Participant arranges puzzle designs using as few moves as possible. Assesses executive planning proficiency and problem-solving. Outcomes include initiation time, execution time, total time of completion, time/rule violations, and a total achievement score. |

ml). Manufacturer's instructions were followed for all assays. Duplicate samples were quantified using standard curves based on calibrators of known concentration; intra-assay % CVs ≤ 5.2% for all assays.

## Data analysis

Statistical analysis was conducted using SPSS Statistics version 29 (IBM Corp, Armonk, NY), with a two-sided p value < .05 deemed statistically significant. Due to the small sample size, all comparisons between the long COVID and asymptomatic control groups on self-report measures and neurocognitive tests were conducted using non-parametric tests, specifically, Chi square or Fishers' exact tests for discrete variables and Mann-Whitney U-tests for continuous variables. Group comparisons for biomarker values utilized the natural log (Ln) for all biomarkers and were conducted using independent samples T-tests. Since the study is based on relatively few participants and there are no published data on recommended cut-off levels for biomarkers applicable to PCC patients, additional analyses were conducted by dichotomizing the biomarkers into below the median vs equal to or above the median based on aggregate data for all participants. Using these cutoffs, group comparisons were conducted to examine whether the proportion of participants across the two groups above and below the median differed statistically. In a final step, a Pro-inflammatory Index was created by summing up the absolute values for IL-10 and IL6. The long COVID scores were compared with the controls using total

scores as well as the Ln transformed scores. The serum Pro-inflammatory Index was dichotomized into less than the median (=5.4005 pg/mL) vs greater than or equal to the median. Fisher's exact 2-sided test was used for these analyses, with significance set at p < .05.

### Ethical considerations

The study was approved by the Institutional Review Board at Corewell Health (IRB nr. 2020–601).

## Results

The sample was comprised of a total of 17 participants who had previously contracted COVID-19, 10 with long COVID and 7 controls. Demographic characteristics of study participants are summarized and compared by group in Table 3. The mean age was 42 years in the long COVID group and 44.32 in the controls. All but one participant was female, all but two identified as White, and none identified as Hispanic. Most participants in both groups had a high school diploma or associate degree as their highest educational degree. All seven control group participants were employed, compared to 6 of 10 in the long COVID group. Three of the 4 unemployed reported that their unemployment was due to their post COVID condition. None of the group differences were statistically significant.

### Self-report measures and neurocognitive tests

Group differences on self-report measures for emotional functioning/quality of life and mood, and on objective neuropsychological tests are summarized in Table 4. On self-report measures, individuals with long COVID scored significantly higher than controls, indicating a higher level of problems on the quality of life measures for usual activity (2.20 vs. 1.00, p = .002) and pain/discomfort (2.20 vs. 1.17, p = .007), as well as on the EuroQol 5-Dimension (EQ-5D EQ5D3L) total score (9.50 vs. 5.50, p < .001). There were no significant group differences for mobility, self-care, and anxiety/depression. The long COVID group scored significantly lower (mean 38.10) than the non-symptomatic group (mean 89.17) on the

**Table 3. Demographics of Long COVID (n = 10) vs Controls (n = 7).**

|  | Long COVID | Controls | p |
|---|---|---|---|
| Age years (SE) | 42.00 (3.18) | 44.32 (4.89) | ns |
|  | N (%) | N (%) |  |
| Gender |  |  | ns |
| Male | 1 (10) | 0 |  |
| Female | 9 (90) | 7 (100) |  |
| Race |  |  | ns |
| Black/African American | 0 | 1 (14.3) |  |
| White/Caucasian | 10 (100) | 5 (71.4) |  |
| Other | 0 | 1 (14.3) |  |
| Highest educational degree |  |  | ns |
| High school diploma | 5 (50) | 4 (57.1) |  |
| Associate's | 2 (20) | 2 (28.6) |  |
| Bachelor's | 3 (30) | 0 |  |
| Master's | 0 | 1 (14.3) |  |
| Employment status |  |  | ns |
| Employed | 6 (60) | 7 (100) |  |
| Unemployed | 4 (40) | 0 |  |

SE, standard error; significance tested using Mann-Whitney U-tests and Chi square statistics.

**Table 4. Self-report measures and neurocognitive testing of Long COVID (n = 10) vs Controls (n = 7).**

| | Long COVID | Controls | |
|---|---|---|---|
| | mean | mean | *p* |
| *Quality of Life: EuroQol 5-Dimension (EQ-5D)* | | | |
| Mobility | 1.60 | 1.00 | ns |
| Self-Care | 1.40 | 1.00 | ns |
| Usual Activity | 2.20 | 1.00 | .002 |
| Pain/Discomfort | 2.20 | 1.17 | .007 |
| Anxiety/Depression | 2.10 | 1.33 | ns |
| VASRaw | 38.10 | 89.17 | .005 |
| EQ5D3L Raw | 9.50 | 5.50 | <.001 |
| *Quality of Life: SF-36* | | | |
| Phys Funct Raw | 46.00 | 97.50 | .003 |
| Role Funct Phys Raw | 10.00 | 100.00 | .002 |
| Role funct Emo Raw | 9.93 | 100.00 | <.001 |
| Energy Fatigue Raw | 17.00 | 70.00 | <.001 |
| Emo Well-Being Raw | 48.80 | 82.67 | .003 |
| Social Funct Raw | 21.25 | 93.75 | <.001 |
| Pain Raw | 45.25 | 96.25 | .003 |
| Gen Health Raw | 28.00 | 81.67 | <.001 |
| *Mood* | | | |
| Beck Depression Raw | 27.50 | 3.86 | <.001 |
| Beck Anxiety Raw | 21.40 | 4.57 | .002 |
| *Learning and Memory: California Verbal Learning Test (CVLT-II)* | | | |
| Cali Verbal 1–5 Learning Total Raw | 46.30 | 53.43 | ns |
| Cali Verbal 1–5 Learning Total T Score | 45.70 | 52.29 | ns |
| Cali Verbal Learning Trial 1 Raw | 5.90 | 6.43 | ns |
| Cali Verbal Learning Trial 1 Z-Score | -0.65 | -0.36 | ns |
| Cali Verbal Learning Trial 5 Raw | 11.40 | 13.14 | ns |
| Cali Verbal Learning Trial 5 Z-Score | -0.60 | -0.07 | ns |
| Cali Verbal Learning List B Raw | 5.00 | 5.43 | ns |
| Cali Verbal Learning List B Z-Score | -0.75 | -0.50 | ns |
| Cali Verbal Test 1–5 Learning Slope Raw | 1.38 | 1.69 | ns |
| Cali Verbal Test 1–5 Learning Slope Z-Score | -0.10 | 0.36 | ns |
| Cali Verbal Test Short Delay Free Recall Raw | 10.60 | 11.00 | ns |
| Cali Verbal Test Short Delay Free Recall Z-Score | -0.20 | -0.14 | ns |
| Cali Verbal Test Short Delay Cued Recall Raw | 11.90 | 11.86 | ns |
| Cali Verbal Test Short Delay Cued Recall Z-Score | -0.15 | -0.21 | ns |
| Cali Verbal Test Long Delay Free Recall Raw | 10.20 | 12.00 | ns |
| Cali Verbal Test Long Delay Free Recall Z-Score | -0.45 | 0.07 | ns |
| Cali Verbal Test Long Delay Cued Recall Raw | 11.40 | 12.29 | ns |
| Cali Verbal Test Long Delay Cued Recall Z-Score | -0.45 | -0.21 | ns |
| Cali Verbal Test Recognition Hits Raw | 13.50 | 15.14 | ns |
| Cali Verbal Test Recognition Hits Z Score | -1.20 | -0.29 | ns |
| Cali Verbal Test False Pos Raw | 2.00 | 0.71 | ns |
| Cali Verbal Test False Pos Z Score | 0.10 | -0.43 | ns |
| Cali Verbal Test Recognition Discrimination Raw | 2.87 | 3.53 | ns |

*(Continued)*

**Table 4.** (Continued)

| | Long COVID | Controls | |
|---|---|---|---|
| | mean | mean | *p* |
| Cali Verbal Test Recognition Discrimination ZScore | -0.35 | 0.50 | ns |
| *Attention: Wechsler Adult Intelligence Scale Digit Span (WAIS-IV Digit Span)* | | | |
| Wechsler Digit Forward Raw | 9.70 | 10.14 | ns |
| Wechsler Digit Backward Raw | 7.70 | 6.86 | ns |
| Reliable Digit Span Raw | 9.00 | 8.57 | ns |
| Wechsler Seq Span Raw | 6.70 | 8.00 | ns |
| Wechsler Digit Span Scaled Score | 8.63 | 8.71 | ns |
| *Processing Speed: Symbol Digits Modality Test* | | | |
| Symbol Digit Mod Raw | 43.50 | 50.43 | ns |
| Symbol Digit Mod Z Score | -0.99 | 0.17 | ns |
| *Executive Functioning: Trail Making Test* | | | |
| Trail Making Part A Raw | 31.60 | 22.86 | ns |
| Trail Making Part A T Score | 45.10 | 56.57 | ns |
| Trail Making Part A Errors Score | 0.33 | 0.40 | ns |
| Trail Making Part B Raw | 94.12 | 69.00 | ns |
| Trail Making Part B T Score | 40.20 | 51.14 | ns |
| Trail Making Part B Errors Score | 0.56 | 0.20 | ns |
| *Executive Functioning: Tower of London (TOL-2)* | | | |
| London Drexel Total Move Raw | 32.67 | 36.00 | ns |
| London Drexel Total Move Standard Score | 98.89 | 96.00 | ns |
| London Drexel Total Correct Raw | 3.78 | 4.00 | ns |
| London Drexel Total Correct Standard Score | 97.56 | 98.57 | ns |
| London Drexel Total Rule Viol Raw Score | 0.33 | 0.00 | ns |
| London Drexel Total Rule Viol Standard Score | 94.67 | 104.29 | ns |
| London Drexel Total Time Viol Raw | 0.89 | 0.86 | ns |
| London Drexel Total Time Viol Standard Score | 93.78 | 93.14 | ns |
| London Drexel Total Initiation Time Raw | 78.00 | 51.29 | ns |
| London Drexel Total Initiation Standard Score | 108.89 | 100.00 | ns |
| London Drexel Total Execution Raw | 220.22 | 207.86 | ns |
| London Drexel Total Execution Standard Score | 97.78 | 98.29 | ns |
| London Drexel Total Prob Solv Raw | 298.22 | 265.43 | ns |
| London Drexel Total Prob Solv Standard Score | 94.89 | 98.86 | ns |
| *Language: Controlled Oral Word Assoc. Test (COWAT)* | | | |
| Control Oral Assoc Letter Fluency Raw | 26.80 | 38.00 | .043 |
| Control Oral Assoc Letter Fluency T Score | 34.10 | 46.71 | .019 |
| Control Oral Assoc Semantic Fluency Raw | 16.20 | 18.14 | ns |
| Control Oral Assoc Semantic Fluency T Score | 34.80 | 45.00 | ns |
| *Premorbid Intellect: Test of Premorbid Functioning* | | | |
| Premorbid Funct Raw | 36.20 | 28.14 | ns |
| Premorbid Funct T Score | 93.80 | 86.57 | ns |
| *Visual Planning: Rey-Osterrieth Complex Figure Test* | | | |
| Rey Osterrieth Raw | 28.50 | 29.86 | ns |
| Rey Osterrieth T Score | -0.49 | -0.69 | ns |

Note: Significance tested using Mann-Whitney U tests; ns, non-significant.

visual analogue scale (VAS) for self-rated health, ranging from 0 (worst health you can imagine) to 100 (best health you can imagine). On the SF-36 quality of life measures, the long COVID group scored significantly lower than the healthy controls on all eight dimensions (p < .01). Thus, those with post-COVID cognitive complaints experienced greater limitations in physical and social activities due to physical and emotional health problems, and reported worse bodily pain, general mental health, and lower vitality, i.e., less energy and more fatigue, than controls. The long COVID group also scored significantly higher on both mood measures (depression, mean 27.50 vs. 3.86, p < .001; anxiety, 21.40 vs. 4.57, p = .002) than healthy controls.

With regard to objective neuropsychological test measures, there were no statistically significant group differences detected for premorbid functioning, learning and memory, attention, processing speed, executive functioning, or visual planning. By contrast, the long COVID group scored significantly lower on two measures of language related to letter fluency (raw score, 26.80 vs. 38.00, p = .043; T-Score 34.10 vs. 46.71, p = .019, Table 4).

## Blood Biomarkers

Comparison of mean biomarker values between groups is presented in Table 5. The only significant difference between groups was for serum NGF levels, which was significantly lower in the long COVID group (mean 9.72) compared to the control group (13.52, p = .038). Biomarker values for TNF-alpha and pTau 181 were not detectable in the study sample and are therefore not reported. Values for pTau 217 were detected in only two individuals, one from each group. In both plasma and serum, levels of pTau 217 were higher in the Long COVID individual (mean plasma .8100 vs .1900; mean serum 1.01 vs .4700).

When comparing groups based on biomarker scores below and above the median, the long COVID group was significantly more likely than controls to have serum IL-10 values ≥ the median (p = 0.015). Eight out of ten participants in the long COVID group were classified into the ≥ median group vs 1 of 7 in the control group. The increased inflammatory drive in long COVID patients was confirmed in terms of the combined Pro-inflammatory Index, which included IL-6 as well

**Table 5. Biomarkers of Long COVID (n = 10) vs Controls (n = 7).**

|  | Long COVID | Controls |  |
| --- | --- | --- | --- |
|  | mean* | mean | p |
| Cortisol Plasma | 81.38 | 64.52 | ns |
| Cortisol Serum | 70.09 | 61.29 | ns |
| DHEA-S Plasma | 6058.83 | 1943.27 | ns |
| DHEA-S Serum | 3227.34 | 2773.22 | ns |
| IL6 Plasma | 1.25 | 1.29 | ns |
| IL6 Serum | 1.80 | 0.84 | ns |
| IL10 Plasma | 2.00 | 2.20 | ns |
| IL10 Serum | 4.45 | 2.96 | ns |
| BDNF Plasma | 897.45 | 779.73 | ns |
| BDNF Serum | 1502.62 | 1635.18 | ns |
| NGF Plasma | 27.24 | 25.98 | ns |
| NGF Serum | 9.72 | 13.52 | .038 |
| NfL Plasma | 7.52 | 12.91 | ns |
| NfL Serum | 10.33 | 13.36 | ns |

Note: Significance tested using independent samples t tests and log-transformed biomarkers; ns, non-significant; DHEA-S, Dehydroepiandrosterone sulfate; IL, interleukin; BDNF, brain-derived neurotrophic factor; NGF, nerve growth factor; NfL, neurofilament light chain; * units for cortisol and DHEA-S, ng/ml; all other biomarkers, pg/ml.

as IL-10. Nine out of 10 patients in the long COVID group exhibited serum pro-inflammatory index values ≥ the median compared to one out of seven in the control group (Fisher's exact test, p = 0.042). For all other biomarkers, there were no significant differences in terms of distribution of variables into below vs equal to and above the median.

## Discussion

This exploratory pilot study compared two groups of laboratory-confirmed post-COVID patients at least 6 months after having been diagnosed with laboratory-confirmed COVID-19. One group was comprised of individuals who had fully recovered (controls) and the other those continuing to experience cognitive difficulties, referred to here as the long COVID group. To the best of our knowledge, this study of cognitive symptoms was the first to use a study design where both the PCC (long COVID) and control groups had a history of PCR-verified COVID-19 infection. Previous studies on cognitive impairment defined healthy controls as individuals who had not had laboratory-confirmed COVID [7,15,26,29]. In an effort to understand why long COVID patients experience prolonged cognitive difficulties such as brain fog, the current study aimed to compare self-reported and objective measures in PCC patients with cognitive complaints with fully recovered COVID patients. The overall study objective was to inform the current understanding of neuropathophysiological disease mechanisms contributing to non-recovery from COVID-19. Moreover, we were interested in findings that might inform the clinical management of PCC.

From a cognitive standpoint, no significant differences were seen between the two groups as it pertained to educational attainment or reading ability which is the most commonly used premorbid estimate for neuropsychological purposes. Given comparable premorbid estimates, with the exception of the letter fluency scores on the Controlled Oral Word Association Test (COWAT), this study found no significant differences between groups in the neurocognitive test battery. This is in line with previous research that found normal cognitive test results in adults with post-COVID cognitive symptoms [14,16]. By contrast, when dividing the PCC group into those with and without cognitive complaints, Ariza et al. [17] found that those with cognitive complaints scored significantly worse on global cognition, learning, memory, processing speed, language, and executive function. However, although our findings were isolated to COWAT, this finding per se is novel since it has never been reported before, and the finding supports PCC patients' self-reports of cognitive inefficiency and perhaps COVID fog. Specifically, this task measures one's ability to engage in higher level thinking (a language task) under time constraint. This could help to explain patients' reports of being able to cognitively engage but feeling that they are foggy and that things take longer and do not seem as automatic.

More consistent group differences were found on self-report measures assessing functional status, quality of life and mood. Specifically, the long COVID group in this study scored significantly and consistently worse than controls on most validated self-report measures, indicating lower ratings on quality of life, physical health, emotional functioning, and psychological well-being. Ariza et al. [15] also reported lower quality of life and functioning scores among post-COVID patients compared to healthy controls. However, unlike the present study that focused only on PCC patients with cognitive complaints, the study by Ariza et al. [15] encompassed patients with a variety of post-COVID complaints. Moreover, the control group in that study had not had a COVID-19 infection. By contrast, controls in the current study were fully recovered COVID-19-infected patients who had not experienced any post-COVID symptoms.

With regard to blood biomarkers, the current study presents intriguing evidence in terms of decreased neuroplastic functioning in PCC patients vs the control group, as reflected in decreased serum levels of nerve growth factor (NGF). NGF is a neurotrophic protein that helps regulate neuronal and neurite growth [56] as well as neuronal phenotypic maintenance and immune function [57]. Brain plasticity and neuro-immunological functioning are closely related. Our findings suggest an attenuation of brain neuroplastic activity. In depressed patients, decreased neuroplasticity has been associated with a range of cognitive difficulties, such as decreased cognitive flexibility, attention, and concentration [58]. In the cognitive aging process, reduced neuroplasticity has also been associated with deterioration of executive functioning, such as working memory and attentional processing [59]. However, the relationship between the two is

complex, and in terms of PCC likely affected by far-reaching effects from neuroinflammation [60]. Such findings might contribute to the numerous patient-perceived reports of attenuated neurocognitive functioning, although possibly not severe enough to be reflected in a systematic decrease in objective neuropsychiatric test scores. Of note, self-rated health is a much better predictor of future health and morbidity status than any clinical sign or laboratory test [61,62]. It might be that the body's internal sensor system is more sensitive and advanced than our current arsenal of clinical and laboratory testing.

However, when comparing groups based on biomarker scores below and above the median, the long COVID group was significantly more likely than controls to have serum values ≥ the median for both IL-10 alone as well as for the Inflammatory Index of IL-6 and IL-10 combined. Lai et al. [22] identified interleukin (IL)-6 as a potential indicator of long COVID, while Queiroz et al. [23] found that patients with no post-COVID symptoms had higher levels of IL-4, IL-6 and IL-10. The discordance on IL-6 in those two studies is difficult to explain but it should be noted that Lai et al [22] conducted a systematic review of 28 studies, while Queiroz et al [23] based their findings on a single cohort of 317 patients with varying degrees of COVID. Queiroz et al [23] did find higher levels of IL-6 among a subgroup of patients with current acute severe infection. Lai et al.'s review [22] covered patients that were in different phases of the COVID infection course. IL-6 is a potent inducer of the acute phase response, and thus plays an important role in the initial phase of an infection. Patients in the later or recovery phase of a COVID infection are not expected to have as high levels of IL-6 as during the initial phase, unless it is a case of PCC [21,22]. In contrast, Queiroz et al [23] followed only one cohort and reported results from a single cross-sectional analysis. In addition, Queiroz et al [23] state that differences in molecular profiles across studies might be due to differences in analytical methodology, infectious background, and other determinants rarely measured in early COVID studies. Such possible differences across studies were not accounted for in either study. However, neither of those studies focused specifically on long COVID patients with cognitive complaints. Two studies that did study long COVID patients with cognitive complaints found no cytokines or inflammatory markers associated with cognitive performance [25,26]. Another common limitation in the above cited work is the lack of objective verification of COVID-19 infection history. This limits the ability to determine whether findings are related to COVID-19 infection history specifically or due to other possible confounders.

When inflammatory marker scores were grouped above and below the median, a larger proportion of long COVID patients fell into the ≥ median group for IL-10. IL-10 represents both a pro- and anti-inflammatory marker [63]. In the current study, we used it as a pro-inflammatory marker. Prior research has also suggested that IL-10 is related to self-reported energy [64]. In the latter case, our findings might suggest a compensatory mechanism by which the body is trying to increase energy to counter the low energy and high fatigue symptoms reported by a majority of long COVID patients. These findings of ≥ median levels of IL-6 and IL-10 support the hypothesis that there is residual activation of the inflammatory systems in the long COVID group versus those that have recovered fully from PCR-verified COVID-19. Looking at the proportion of participants falling into the high vs. low serum pro-inflammatory group, all except one person in the long COVID group fell into the high serum pro-inflammatory group vs one out of seven in the control group. Thus, overall, using several different definitions of pro-inflammation, the long COVID group systematically showed a pattern of heightened inflammation as compared to the control group that has fully recovered from their COVID-19 infection.

## Limitations

These results should be viewed in consideration of several study limitations. First and foremost is the small sample size, which makes generalizability to other populations difficult, although we used a rigorous assessment scheme. Small sample sizes have reduced statistical power to detect true effects and results may be affected by outliers. We were also not able to recruit 15 participants in either group, with recruitment of controls especially difficult. However, we saw a consistent pattern of lower scores on all self-rated health and quality of life measures, and higher depression and anxiety scores in

the long COVID group, which serves as validation for the group differences seen on biomarkers when examining values above and below median scores. The study sample was not diverse, as participants were primarily female and White. Notably, this was an exploratory pilot study that was, to the best of our knowledge, unique in the comparison of COVID survivors with and without post-illness cognitive symptoms. Thus, our control group had lab-verified histories of COVID infection from which they had fully recovered.

## Implications for clinical care

Results of this pilot study point to few but possibly clinically important differences between long COVID and control patients with regard to neurocognitive tests, but substantial differences related to physical and emotional functioning and quality of life. Although this was a small sample, results also suggest possible prolonged inflammation in the long COVID group, which has been suggested previously [27], however, without using a proper reference group. An interesting observation is the parallel between this disorder and chronic fatigue syndrome (CFS), as both have patients reporting mood issues such as anxiety [65]. This raises the question of what we can learn from the experience of patients with CFS and whether that can be applied to long COVID patients with cognitive complaints. This may include consideration of the types of clinical teams necessary to support the patients and their caregivers as well as treatment. For example, there is no standard of care when it comes to treatment for CFS. Anhydrous Enol-Oxaloacetate (AEO) is a nutritional supplement which has anecdotally been reported to relieve physical and mental fatigue in Myalgic Encephalomyelitis/Chronic Fatigue Syndrome (ME/CFS) and long COVID patients [66]. Could this be a direction for us to consider for quality-of-life outcomes for long COVID patients as well? Another factor to consider is the resources needed to create longitudinal and multidisciplinary interventions that can support patients long term - encompassing perceptions of quality of life, physical functioning, emotional health and wellbeing and vocational rehabilitation - and the cost associated with such efforts. Combined exercise and behavioral support, developed with extensive patient and stakeholder engagement, is being tested in a randomized controlled trial in the United Kingdom [67]. Should it prove to be clinically cost-effective for people with long COVID, there is an opportunity to create similar programs.

In addition to the above, our results also support the importance of considering neuroinflammatory processes as treatment to this point has mostly focused on behavioral and cognitive interventions, assuming there are no residual neuroinflammatory processes or deficits. This points to the importance of applying a multidisciplinary approach in addressing long COVID patients, including a rigorous assessment to detect possible residual systemic inflammation and reduced neuroplasticity. Such an approach could help to identify which patients may transition to experiencing long COVID and is likely to expand the arsenal of choices of promising and evidence-based treatment strategies with the ultimate aim to design rigorous clinical trials.

## Conclusions

Long COVID patients with cognitive complaints experience significantly more anxiety and depression, lower self-rated health, and lower physical and emotional functioning and quality of life compared to fully recovered COVID-19 patients, although only one neurocognitive test differed between groups. Biomarker analyses suggest possible prolonged inflammatory processes in long COVID patients. Moreover, results of decreased neuroplastic functioning give credence to patients' reports of changes in brain function. Future studies in larger, more diverse samples are required to fully understand these differences and to develop effective clinical treatments for those with cognitive difficulties.

## Supporting Information

### S1 Dataset. Dataset.

(XLSX)

## Acknowledgments

The authors extend their gratitude to the patients who participated in the study and to John Beck, BS for conducting the biomarker analysis.

## Author contributions

**Conceptualization:** Michael R. Lawrence, Judith E. Arnetz, Aiesha Ahmed, Bengt B. Arnetz.

**Data curation:** Michael R. Lawrence, Bengt B. Arnetz.

**Formal analysis:** Judith E. Arnetz, Scott E. Counts, Bengt B. Arnetz.

**Funding acquisition:** Aiesha Ahmed, Bengt B. Arnetz.

**Investigation:** Michael R. Lawrence.

**Methodology:** Michael R. Lawrence, Judith E. Arnetz, Aiesha Ahmed, Bengt B. Arnetz.

**Project administration:** Michael R. Lawrence, Bengt B. Arnetz.

**Resources:** Michael R. Lawrence, Scott E. Counts, Bengt B. Arnetz.

**Software:** Judith E. Arnetz, Bengt B. Arnetz.

**Supervision:** Bengt B. Arnetz.

**Validation:** Michael R. Lawrence, Judith E. Arnetz, Scott E. Counts, Bengt B. Arnetz.

**Visualization:** Michael R. Lawrence, Judith E. Arnetz, Bengt B. Arnetz.

**Writing – original draft:** Michael R. Lawrence, Judith E. Arnetz, Aiesha Ahmed, Bengt B. Arnetz.

**Writing – review & editing:** Michael R. Lawrence, Judith E. Arnetz, Scott E. Counts, Aiesha Ahmed, Bengt B. Arnetz.

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
