## [Decision Letter · Decision Letter 0]

4 Mar 2025

PONE-D-24-53401Self-reported health, neuropsychological tests and biomarkers in fully recovered COVID-19 patients vs patients with post-COVID cognitive symptoms: a pilot studyPLOS ONE

Dear Dr. Arnetz,

Thank you for submitting your manuscript to PLOS ONE. After careful consideration, we feel that it has merit but does not fully meet PLOS ONE’s publication criteria as it currently stands. Therefore, we invite you to submit a revised version of the manuscript that addresses the points raised during the review process.

We look forward to receiving your revised manuscript.

Kind regards,

Zypher Jude G. Regencia, Ph.D.

Academic Editor

PLOS ONE

Journal Requirements:

“This study was financially supported by a grant from the Spectrum Health Foundation, principal investigators AA and BBA. The funders did not play any role in the study design, data collection and analysis, decision to publish, or preparation of the manuscript.”

“The authors extend their gratitude to the patients who participated in the study and to John Beck, BS for conducting the biomarker analysis. This study was financially supported by a grant from the Spectrum Health Foundation (Grant nr. FDN 35911-2020-601).”

“This study was financially supported by a grant from the Spectrum Health Foundation, principal investigators AA and BBA. The funders did not play any role in the study design, data collection and analysis, decision to publish, or preparation of the manuscript.”

Reviewers' comments:

Reviewer's Responses to Questions

**Comments to the Author**

1. Is the manuscript technically sound, and do the data support the conclusions?

Reviewer #1: Yes

Reviewer #2: Yes

2. Has the statistical analysis been performed appropriately and rigorously? 

Reviewer #1: Yes

Reviewer #2: Yes

3. Have the authors made all data underlying the findings in their manuscript fully available?

Reviewer #1: Yes

Reviewer #2: No

4. Is the manuscript presented in an intelligible fashion and written in standard English?

Reviewer #1: Yes

Reviewer #2: Yes

5. Review Comments to the Author

Reviewer #1: The authors present a very insightful pilot study on the difference in mood, quality of life, and neuro-cognitive scores, and biomarkers between COVID-19 survivors with persistent cognitive symptoms and fully-recovered COVID-19 patients using the case-control design. The pilot study contributes additional exploration on the post-COVID-19 phenomenon to the existing published studies by recruiting laboratory-confirmed COVID-19 survivors as controls and by including both cognitive batteries and biomarkers as outcome measures. I do have some minor recommendations for the authors:

1. The difference between the ‘fully recovered COVID-19 patients’ and ‘patients with post-COVID cognitive symptoms’ groups is very arbitrary and highly reliant on single self-report. The authors are enjoined to specify whether they use the United Kingdom National Institute for Health and Care Excellence clinical case definition of post-COVID-19 condition or the World Health Organization 2021 Delphi Study case definition (accessible via https://iris.who.int/bitstream/handle/10665/345824/WHO-2019-nCoV-Post-COVID-19-condition-Clinical-case-definition-2021.1-eng.pdf?sequence=1 ).

2. It would be ideal to relocate the statement on line 130 “The sample was comprised of a total of 17…” to the Results section. The authors are also encouraged to at least declare if there was an attempt to reach a specific sample size for the pilot study involving continuous variables- the flat rule or the stepped rule of sample size determination for pilot studies.

3. It would be prudent to state why the other biomarkers mentioned in the review - GFAP, IL 12, IL 17, CRP, and TNF alpha - were not included in the pilot study.

4. The authors are enjoined to use the Vancouver citation style wherein the in-text citation entails a superscript number after the period in compliance with the PLOS ONE submission guidelines.

Reviewer #2: The study fills important knowledge gaps in the elucidation of COVID-19 as a disease. The novelty this study brings is the rigorous comparison between prior lab-confirmed COVID-19 patients that have recovered (i.e., asymptomatic) versus lab-confirmed COVID-19 patients that have developed cognitive symptoms at least 6 months since diagnosis. The comparison spans cognitive and psychological function, perceptions of functional status and quality of life, and serum biomarkers of stress, inflammation, and neuroplasticity. The main weakness is the limited sample size.

Major

- In lines 74–76, the authors talk about post-COVID cognitive symptoms having higher frequency among individuals who had mild infection and who have a history of cognitive difficulties and depressive disorder. However, in the methodology, the authors do not include information on whether the cohort they recruited did have these factors. Knowing this clinical information is important as these conditions could confound the analysis. For example, a person who might have history of cognitive difficulties may fare worse in the neurocognitive function tests. How about vaccination history?

- For study participants, did the authors set any exclusion criteria? Since the analysis assesses cognitive and psychological functioning, quality of life, and serum biomarkers, confounders such as history of neurological and psychological conditions, any disabilities, or any immunological conditions might impact the data.

Minor

- In the introduction (line 56), the authors define Post-COVID conditions (PCC) as a syndrome characterized by symptoms that extend well beyond the initial recovery period. The statement sounds arbitrary. Although the definition has not been standardized yet, perhaps the authors can include the usual time duration of recovery to make the statement less abstract and more clear.

- How is “fully recovered patients” qualified? Since assessing recovery in this study relies on self report of symptoms, it is important to enumerate the review of systems asked (even just as supplemental information) so readers might be able to assess how comprehensive the assessment was.

- Lines 111-113: What could be the reason why in ref 15, patients with PCC cognitive complaints did not differ from healthy controls on neurocognitive testing but tested WORSE than PCC without cognitive complaints?

- The authors did not indicate the details on blinding and whether it was done in this study.

- Lines 290-291: The authors should be careful with the use of the word “sustained” to describe the attenuation in neuroplastic activity especially since the data is not longitudinal.

- Lines 291–292: What is the link between neuroplasticity and cognition? Is a decrease in neuroplasticity linked with cognitive problems? Which specific aspect of cognition has established links in literature? Perhaps the authors can include a concise statement on this.

- Line 301: In the discussion by the authors, what may explain the discordance on IL-6 findings between the Lai et al. and Queiroz et al. studies?

- Lines 164–165: Consider reviewing the sentence for typographical errors.

- Line 191: Consider reviewing the case of the phrase “pro-inflammatory Index” (common vs. proper noun; consistency throughout the text)

- Lines 254-255: Consider reviewing the sentence, particularly on parallelism (“examine and compare self-reported and objective measures in PCC patients with cognitive complaints with…”)

- Line 345-347: Consider putting the comma after the initialism AEO. Also consider reviewing the sentence. The authors are requested to clarify the sentence: Is it the AEO that is diminished in CFS patients?

- Line 349-351: Consider using an em dash instead of a hyphen.

- Line 354: The authors should be consistent in their use of the phrase “long COVID” and not hyphenate it. Also in line 358: “neuroinflammatory”

6. PLOS authors have the option to publish the peer review history of their article (what does this mean? ). If published, this will include your full peer review and any attached files.

**Do you want your identity to be public for this peer review?** For information about this choice, including consent withdrawal, please see our Privacy Policy .

Reviewer #1: No

Reviewer #2: **Yes: ** Ben Anthony A. Lopez, MD, PhD

---

## [Author Response · Author response to Decision Letter 1]

23 Mar 2025

Response to reviewer comments are summarized in a point-by-point document, "Response to reviewer comments" attached to this submission.

---

## [Editor Report · Decision Letter 1]

27 Mar 2025

Self-reported health, neuropsychological tests and biomarkers in fully recovered COVID-19 patients vs patients with post-COVID cognitive symptoms: a pilot study

PONE-D-24-53401R1

Dear Dr. Arnetz,

We’re pleased to inform you that your manuscript has been judged scientifically suitable for publication and will be formally accepted for publication once it meets all outstanding technical requirements.

Kind regards,

Zypher Jude G. Regencia, Ph.D.

Academic Editor

PLOS ONE
---

## [Editor Report · Acceptance letter]

PONE-D-24-53401R1

PLOS ONE

Dear Dr. Arnetz,

I'm pleased to inform you that your manuscript has been deemed suitable for publication in PLOS ONE. Congratulations! Your manuscript is now being handed over to our production team.

Kind regards,

on behalf of

Dr. Zypher Jude G. Regencia

Academic Editor

PLOS ONE